# Nonlinear Measures to Evaluate Upright Postural Stability: A Systematic Review

**DOI:** 10.3390/e22121357

**Published:** 2020-11-30

**Authors:** Justyna Kędziorek, Michalina Błażkiewicz

**Affiliations:** Akademia Wychowania Fizycznego Józefa Piłsudskiego w Warszawie, 00-809 Warszawa, Poland; justyna.kedziorek@awf.edu.pl

**Keywords:** sample entropy, fractal dimension, Lyapunov exponent, center of pressure

## Abstract

Conventional biomechanical analyses of human movement have been generally derived from linear mathematics. While these methods can be useful in many situations, they fail to describe the behavior of the human body systems that are predominately nonlinear. For this reason, nonlinear analyses have become more prevalent in recent literature. These analytical techniques are typically investigated using concepts related to variability, stability, complexity, and adaptability. This review aims to investigate the application of nonlinear metrics to assess postural stability. A systematic review was conducted of papers published from 2009 to 2019. Databases searched were PubMed, Google Scholar, Science-Direct and EBSCO. The main inclusion consisted of: Sample entropy, fractal dimension, Lyapunov exponent used as nonlinear measures, and assessment of the variability of the center of pressure during standing using force plate. Following screening, 43 articles out of the initial 1100 were reviewed including 33 articles on sample entropy, 10 articles on fractal dimension, and 4 papers on the Lyapunov exponent. This systematic study shows the reductions in postural regularity related to aging and the disease or injures in the adaptive capabilities of the movement system and how the predictability changes with different task constraints.

## 1. Introduction

Postural control is a term used to describe how the central nervous system regulates sensory information from other systems to produce adequate motor output to maintain a controlled upright posture. Postural control is a complicated phenomenon that combines both postural orientation and postural equilibrium. Postural orientation involves the active alignment of the trunk and head in relation to the line of gravity, the base of support, the visual surround and internal references. Sensory information from somatosensory, vestibular, and visual systems are integrated and the relative weights placed on each of these input data depend on the objectives of the motor task and the environmental context [1]. Postural equilibrium involves the coordination of movement strategies to stabilize the centre of body mass during both self-initiated and externally induced perturbations of stability. Therefore, the selected specific response strategy depends not only on the characteristics of the external postural displacement but also on the individual expectations, goals, and previous experiences.

The most common technique used to quantify postural control in upright stance is the assessment of the variability of the center of pressure (CoP). However, in recent years wearable sensors, as well as motion capture systems, are becoming increasingly common method for evaluating postural stability. The advantage of these methods is the ability to evaluate the posture stability in 3D [2,3]. However, the information obtained from these methods cannot be unequivocally interpreted from a physiological point of view. The CoP is in fact a measure of whole-body dynamics and thereby represents the sum of various neuro-musculoskeletal components acting at different joint levels. Furthermore, the CoP’s time series is two dimensional. Although the two components of the signal, anterior–posterior and medial–lateral are often analyzed separately: They represent the output of a unique integrated system. As a consequence, the utility of static posturography in clinical practice is somehow limited and there is a need for reliable approaches in order to extract physiologically meaningful information from stabilograms. Therefore, the techniques of CoP signal evaluation have been recently described by using the dynamic approach. Nonlinear measures are capable of capturing the temporal component of the variation in CoP displacement with regard to how motor behavior develops over time. Therefore, these measures allow for quantifying regularity, adaptability to environment, stability [4,5], and complexity [6]. In approach reviewed in this paper, many authors assume that complexity can be defined as a compromise between order and disorder and between simplicity and complication [6]. Thus, complexity is related to the properties of stability and adaptability that characterized healthy systems and which could be lost with aging and disease. Nonlinear tools for evaluating the above-mentioned postural control properties include the largest Lyapunov exponent and Hurst exponent, recurrence quantification analysis (RQA), as well as fractal dimension and entropy families [4,7,8].

Sample entropy (SampEn) is one of the various types of entropy measures. This coefficient is used to determine the regularity of postural sway and quantifies the temporal structure of the signal by calculation the probability of that two similar sequences with the same number of data points remain similar when another data point is added [9]. In other words, SampEn (m, r, N) of a dataset of length N is the negative natural logarithm of the conditional probability of two successive counts of similar pairs (Chebyshev distance less than a tolerance size of r) of template size m and m + 1 without allowing self-matches. Chebyshev distance is also called maximum value distance and it examines the absolute magnitude of the differences between two vectors or points [10]. An advantage of SampEn is the independence of data length [11]. However, Richman and Moorman [11] advised caution when using datasets less than 200 points. The increased values of SampEn indicate larger irregularity of the CoP, which is more random and less predictable. Lower SampEn values show that the CoP signal is more regular and predictable, which is associated with less complexity of structure [12]. As complexity is crucial to the flexibility in adaptation to the surroundings, this lower complexity of physical movement translates into lower flexibility and higher rigidity of postural control [13]. Conversely, higher SampEn, which reflects increased complexity, is interpreted as improved self-organization and an effective strategy in postural control [7].

Fractal dimension (FD) is another measure that indicates the complexity of the CoP signal by describing its shape [14]. It shows the complexity and self-similarity of physiological signals. In characterizing the complexity of the CoP path, FD describes the activity of the sensorimotor system in organizing available afferents and the extent to which a person utilizes the base of support available to them [15]. In the peculiar case of the CoP trajectory, a change in FD may indicate a change in control strategies for maintaining a quiet stance. Currently, many algorithms calculate fractal dimension: Higuchi algorithm [16], Maragos and Sun algorithm [17], Katz algorithm [18], Petrosian algorithm [19], and box-counting method [20]. The most appropriate method for calculating the FD for biological signals is the Higuchi algorithm. It does not depend on the binary sequence and in many cases is less sensitive to possible noise [21].

Lyapunov exponent (LyE) is a well-defined tool to characterize the chaotic behavior of the signal. As a nonlinear parameter, this exponent measures the rate of loss of information from chaotic time series. The human dynamic stability characterized by LyE measures the resistance of the human locomotor control system to perturbations [22]. It quantifies how well an individual can keep a stable posture under perturbations in the environment. A higher LyE points to the capability of a more rapid response of balance control in different body movements [23]. In order to facilitate the reading of the general sense of low and high values of nonlinear indices in this review, Table 1 was created. Table 1 provides brief definitions of each coefficients in relation to the assessment of postural control in base on CoP time series.

Different techniques, methods, and various quantitative and qualitative variables measured have been employed in the literature to objectify postural control. Considering that the interest in the dynamic approach has been growing recently, it seems necessary to collect existing data related to the use of chaos indicators to assess postural control. Until now, not many reviews on the use of nonlinear analysis to evaluate postural stability have been found. Most of the manuscripts deal only with individual nonlinear indicators and are not reviews. The reviews which have been found relate to general descriptions of nonlinear measures (mainly approximate and multiscale entropy) and their application or mathematical calculations. Cavanaugh, et al. [27] reviewed the theoretical foundation and limitations of the traditional postural stability model. Following cerebral concussion on athletes without postural instability showed that approximate entropy (ApEn) had detected a subtle change in postural control in the absence of postural instability. Gow, et al. [28] made the systematic review and it has revealed significant heterogeneity in the way Multiscale Entropy Analysis (MSE) is applied to CoP displacement data. Authors highlighted that significant variability in methodological approaches may impact results and their interpretations. They recommend to establish a few factors: The minimal amount of time for data collection, the physiological frequencies to evaluate, the inclusion of healthy controls, sampling rate for data acquisition, way of data filtration, and assigning appropriate values of m—the length of reconstructed vectors (i.e., length of the data segment being compared) and r—the tolerance threshold (i.e., similarity value for comparing reconstructed vectors). The purpose of Busa and van Emmerik [29] paper was to review basic elements and current developments in entropy techniques which had been used to identify how MSE can provide insights into the complication of physiological systems operating at multiple time scales that underlie the control of posture. Authors reviewed the evidence from the literature providing support for MSE as a valuable tool to evaluate the breakdown in the physiological processes that accompany changes due to aging and disease in postural control. This evidence emerged from observed lower MSE values in individuals with multiple sclerosis, idiopathic scoliosis and in older individuals with sensory impairments. At the end, Tang, Lv, Yang, and Yu [25] provided the most comprehensive literature review by examining the various complexity testing techniques for time series data and their application in fields of economics, life science, earth science, engineering, and physics. They distinguished three complexity measures groups: Fractality theory—which focuses on self-similarity and entropy—for the disorder state of a system and methods which explore data dynamics by investigating the strange attractor in phase-space. The authors have operated in a very broad and sophisticated area, showing techniques for counting and interpreting the results. Moreover, they underlined that the above-mentioned groups complexity testing techniques are closely related to or even depend on each other. One year later, van Emmerik, et al. [30] reviewed fundamental concepts of dynamic systems as variability, stability, and complexity of human movement. From this review, it was evident that these important concepts cannot be considered interchangeable and in future research should be distinguished carefully.

In conclusion, the literature lacks a systematic review that shows how the values of selected nonlinear measures in various groups of subjects during the assessment of postural stability in typical free standing tasks looks like and how authors interpret them. For these reasons, the present systematic review aims to investigate the application of nonlinear dynamics coefficients (sample entropy, fractal dimension and Lyapunov exponent) to assess postural stability during upright standing.

## 2. Materials and Methods

### 2.1. Search Strategy

This review was limited to studies in which the nonlinear dynamics coefficients were used to assess postural stability in base of CoP fluctuation. The electronic search of databases was performed in December 2019 by one author JK. The articles were limited to the period from January 2009 to December 2019. PubMed, Google Scholar, Science-Direct, and EBSCO databases were searched to identify appropriate literature using the search terms “postural stability and sample entropy”, “postural stability and fractal dimension”, “postural stability and Lyapunov exponent”, “upright stance and nonlinear measures”, “posturography and nonlinear measures”, and “postural control and nonlinear measures”.

### 2.2. Eligibility

Only full-text articles in English were selected from the electronic databases. The inclusion criteria were: (1) Human participants, (2) focus on postural stability in standing, (3) peer-reviewed full scientific articles, (4) CoP assessment using force plate as a measure of postural stability, (5) sample entropy, fractal dimension, Lyapunov exponent used as nonlinear measures, and (6) availability in the English language. Titles, abstracts and full texts of retrieved documents were sequentially reviewed by authors (JK and MB) to determine their relevance to the topic. Furthermore, the reference lists of all studies included for review were searched manually for additional studies of relevance. Articles that focused on other movements such as gait stability, sitting or stability in the dynamic environment were excluded, as were the papers where measures other than sample entropy, Lyapunov exponent and fractal dimension were used for calculations. The articles in which postural stability was assessed using other devices such as motion capture or accelerometer were also removed. Moreover, the manuscripts which lacked basic information about the equipment or the characteristics of the study group were also excluded. No restriction was applied regarding sex, age, disabilities, injures, or diseases.

### 2.3. Review Process

Duplicate articles from different databases were rejected. The title and abstract for the selected articles were first screened according to the eligibility criteria. Furthermore, the full-text evaluation was performed if the title and abstract could not provide adequate information for the article screening process. Rejected articles were re-screened to avoid misinterpretation. The titles, abstracts and then full text of the papers identified by the search were screened by two independent reviewers (the authors: JK and MB) to choose those that met the selection criteria and extract the data. Decisions about which trials should be selected were made by negotiation. One reviewer (MB) compiled all articles in using a reference manager software (EndNote X7.7, Clarivate Analytics, Philadelphia, PA, USA). Next, the articles that had been found and accepted were divided into three subgroups according to which a nonlinear factor was used to assess postural stability. Three subgroups were distinguished: Sample entropy, fractal dimension, and Lyapunov exponent. Additionally, for each nonlinear parameter, articles describing three types of study groups were distinguished: (1) Children, young and older adults, (2) people with disabilities, injures or diseases, and (3) athletes.

### 2.4. Quality Assessment

The methodological quality of the trials selected for this review was then assessed using a checklist for both of randomized and non-randomized studies [31]. The checklist consisted of 27 items distributed between five sub-scales: (1) Reporting (10 items)—which assessed whether the information provided in the paper was sufficient to allow the reader to make an unbiased assessment of the findings of the study; (2) External validity (3 items)—which addressed the extent to which the findings from the study could be generalized to the population that the study subjects had been derived from; (3) Bias (7 items)—which addressed biases in the measurement of the intervention and the outcome; (4) Confounding (6 items)—which addressed bias in the selection of study subjects; and (5) Power (1 item)—which attempted to assess whether the negative findings from a study could be due to chance. As not all questions were adequate to the analyzed papers, due to high medical bias, only one sub-scale (Reporting) was selected for the evaluation of the works. Reporting the checklist included the following questions: 1. Is the hypothesis/aim/objective of the study clearly described?; 2. Should the main outcomes to be measured and clearly described in the Introduction or Methods section?; 3. Are the characteristics of the patients included in the study clearly described?; 4. Are the interventions of interest clearly described?; 5. Are the descriptions the distributions of principal confounders in each group of subjects clear?; 6. Are the main findings of the study clearly described?; 7. Does the study provide estimates of the random variability in the data for the main outcomes?; 8. Have all the important adverse events that may be a consequence of the intervention been reported?; 9. Have the characteristics of patients lost to follow-up been described?; and 10. Have actual probability values been reported (e.g., 0.035 rather than <0.05) for the main outcomes except where the probability value is less than 0.001? Answers were scored 0 or 1, except item number 5 which scored 0 to 2. The total maximum score was therefore 11.

## 3. Results

Initially, the electronic database screening process yielded 1100 articles for all the parameters. 10 articles were found out to had been duplicated and were removed. Screening of titles and abstracts eliminated 1004 articles, and an agreement was reached for 86 articles, which were identified to be related to the aim of the literature survey. Following the eligibility criterion of full-text studies, 86 articles were selected for the review. Further analysis of the study excluded another 43 full-text records that did not meet the search criteria. A total of 43 articles were selected for the review process (Figure 1).

### 3.1. Sample Entropy

Regarding the use of sample entropy for postural stability evaluation, a total of 510 papers were found in PubMed (5 records), Science-Direct (325 records), EBSCO (176 records), and Google Scholar (4 records). In total, 33 papers were submitted for the analysis (Table 2).

The assessment of postural stability using SampEn in groups of older adults, young people and children was included in 12 papers. Fourteen articles analyzed people with dysfunctions, neurological diseases, and musculoskeletal disorders. Athletes were studied only in 7 papers. Only in 3 papers [40,44,46] the results were analyzed using fractal dimension in addition to sample entropy. In one paper [41] analysis based on sample entropy was supplemented by LyE (Figure 1). All the papers were highly rated (10/11 points). The score was affected by a negative answer to question 8 (Have all important adverse events that may be a consequence of the intervention been reported?) in all of the cases and resulted in losing one point. Only three articles [50,52,55] were rated lower (9/11 points). In this case, a negative answer to question 10 (Have actual probability values been reported (e.g., 0.035 rather than <0.05) for the main outcomes except where the probability value is less than 0.001?) had an impact.

The youngest study group was children aged 3 years (42.3 ± 3.2 months) [42], whereas the oldest group was consisted of adults aged 85.4 ± 4.4 years [37]. Quiet standing trials with eyes open and closed were dominant in all three groups. The duration of each trial ranged from 20 to 120 s for the groups of older adults and young people, from 20 to 60 s in the second group (Disabilities/Injures/Diseases), and 20–30 s in the group of athletes. The CoP sampling rate was in the range of 20 Hz to 1000 Hz, but the most commonly used was 100 Hz. Analyzing the method of SampEn calculation, most of the works did not explain on what basis and how the values of m and r were selected. In 11 papers, default values of m and r parameters (m = 2, r = 0.2) were used [64]. In 13 papers it was not stated whether the SampEn was calculated for the raw or filtered signal.

In the groups of older adults and young children, SampEn was lower for older adults compared to young people [9,33]. In the group Disabilities/Injures/Diseases, the entropy analysis showed lower values in people with injuries, dysfunctions or diseases than those in the group of healthy people. In the group of athletes, the postural sway of dancers/gymnasts was characterized by more irregular CoP sway (as exemplified by higher sample entropy) than in non-dancers. In all of the groups, the absence of vision led to a decrease in SampEn as compared to when the eyes were open. The values of entropy in the analyzed papers are in the range: 0.021 ± 0.009 [37]—1.73 ± 0.1 [48].

### 3.2. Fractal Dimension

Regarding to the use of fractal dimension for postural stability evaluation, a total of 348 papers were found in PubMed (2 records), Science-Direct (94 records), EBSCO (66 records), and Google Scholar (186 records). In total, 10 papers were submitted for analysis (Table 3).

The assessment of postural stability using FD in the groups of older adults and young people was discussed in 4 papers. People with dysfunctions and neurological diseases were analyzed in 5 articles. Athletes were studied only in 1 paper. Only in 3 papers [40,44,46] the authors used SampEn in addition to FD. The same as for articles from Section 3.1, all papers were rated highly (10/11 points). The score was affected by a negative answer to question 8 in all cases and therefore it resulted in losing one point. Only one paper [70] was rated lower (9/11 points). In this case, a negative answer to question 10 had an impact. Analysis of the methods of calculating FD shows that Katz’s algorithm was used in 5 papers, Higuchi’s algorithm appeared in 3 and a box counting method was used in one study. In paper [44], fractal dimension by dispersion analysis based on the standard deviation was used in order to calculate FD. The youngest study group were people aged 20.4 ± 1.8 years [66], whereas the oldest group was aged 69.8 ± 5.6 years [65]. Quiet standing trials with eyes open and closed was common in all three groups. The duration of each trial ranged from 20 to 70 s for older adults and young people and from 20 to 45 s in the group II (Disabilities/Injures/Diseases). The most commonly used sampling rate was 100 Hz, but it ranged from 100 Hz to 500 Hz.

In the groups of older adults and young people, Tassani, Font-Llagunes, Gonzalez Ballester, and Noailly [69] showed a higher value of FD during standing with EO compared to standing with EC. Qiu. H [66] demonstrated that FD was lower in case of older adults compared to young people only in open eyes condition. It is noteworthy that the purpose of Qiu. H [66] work was to compare the test-retest reliability of a wide variety of center of pressure (CoP) based postural sway measures and their ability to detect the differences between the young and older groups, between the older low- and high-fear of falling groups, and between the older non-faller and faller groups. Experimental results showed that FD had acceptable levels of relative and absolute reliability, but was insensitive to detect age-group difference and fear of falling under both visual conditions. In the group of Disabilities/Injures/Diseases, only Cimolin, Galli, Rigoldi, Grugni, Vismara, Mainardi, and Capodaglio [70] found out that individuals with Prader–Willi syndrome were characterized by a greater value of FD compared to the healthy group. Other studies, which are based mainly on the traumatic injuries (ankle sprain, whiplash) showed that FD was higher in groups of healthy people [15,44]. In the group of athletes, Casabona, Leonardi, Aimola, La Grua, Polizzi, Cioni, and Valle [73] observed a higher value of FD in dancers compared to non-dancers. In all of the groups, the absence of vision led to an increase in FD as compared to when the eyes were open. The values of FD in the analyzed papers are in the range: 1.06 ± 0.01 [44]—1.76 ± 0.06 [73].

### 3.3. Lyapunov Exponent

Regarding the use of Lyapunov exponent for postural stability evaluation, a total of 242 papers were found in PubMed (5 records), Science-Direct (171 records), EBSCO (11 records), and Google Scholar (55 records). In total, 4 papers were submitted for analysis (Table 4).

Similar to the papers from Section 3.1 and Section 3.2, all papers were highly rated (10/11 points). The score was affected by a negative answer to question 8 and resulted in losing one point. Only one paper [76] was rated lower (9/11 points). In this case just like mentioned before, a negative answer to question 10 had an impact. In all studies, the Wolf’s method [75] was used to calculate the Lyapunov exponent. In one paper [74], the authors additionally used the multivariate largest Lyapunov exponent method.

The assessment of postural stability using the Lyapunov exponent in the groups of older adults and young people was discussed in 3 papers. Liu, Wang, and Xiao [74] showed that the stability of human standing reduced with age. The LyE value reflected the overall coordination between multi-segment movements and was lower in young individuals. Lower LyE values for the group of younger people were also obtained at work [41]. In the group of people with disabilities, only one study [78] was found and it shows that the stability of patients with multiple sclerosis (PwMS) evaluated during the 5-min standing with eyes open and closed. In PwMS, the CoP time-series showed decreased LyE values compared to controls. The youngest study group were people aged 18.0 ± 0.7 years [79], whereas the oldest group was aged 68.3 ± 2.7 years [41]. The duration of each trial ranged from 20 to 70 s for the groups of older adults and young people. The most commonly used sampling rate was 100 Hz, but it ranged from 100 Hz to 1000 Hz. The values of LyE in the analyzed papers are in the range: 0.12 ± 0.07 [76]—2.23 ± 0.67 [76].

## 4. Discussion

This review aimed to summarize and update information on the current published research explicitly related to the application of nonlinear dynamics coefficients to assess postural stability during upright standing. Following the research on biological signals in various scientific fields, it can be observed that in recent years there has been a trend to analyze the postural stability system using nonlinear measures and consequently, the CoP signal. Conventional parameters are slowly being ousted by nonlinear coefficients, which may be better tools to assess the complex system of balance control. This review examined 43 studies involving sample entropy, fractal dimension, and the Lyapunov exponent in different age groups used to assess static stance in the past 10 years.

A variety of studies have revealed that the variability of CoP time-series during quiet standing is not the result of random error [80,81]. The CoP oscillations despite appearing erratic and irregular, contain a certain orderliness, that emerges in time [27]. In contrast to linear models that analyze the magnitude of output signal (predominantly CoP path length), nonlinear models use the time evolutionary properties of an output signal to draw conclusions about interactions within the control system. Under fixed task and environmental conditions, nonlinear properties of the postural control system arise mainly due to elastic and damping properties of muscles and the varying time scales of sensory systems (such as delays and thresholds) [27,82]. Measures derived from the nonlinear dynamics are based on the recognition that the collective interaction among these properties produces the complex behavior of the postural control system. One of the nonlinear measures, known as sample entropy, quantifies the ensemble amount of randomness, or irregularity. Fractal dimension provides the information about change of the control strategies used for upright balance and has been developed to estimate system complexity in terms of the roughness of time-series. The Lyapunov exponent investigates how the system states change over time in terms of the exponential divergence (or convergence) of initially nearby trajectories, and the growing rate of the separation between nearby trajectories reflects the sensitivity of the system to initial conditions. In particular, it provides a measure of the local stability of a dynamical system. The Lyapunov exponent indicate an ability to adapt to environment (Table 1).

The analysis of the studies describing postural stability using the sample entropy coefficient reveals that some relationships are repeated in the comparisons of the groups of older adults, younger people, people with diseases, disabilities or injures and athletes. Ramdani, Seigle, Varoqui, Bouchara, Blain, and Bernard [33] showed that the absence of visual control in older and younger people translates into lower values of entropy in anterior–posterior (AP) and medial–lateral (ML) directions. Therefore, in both group the regularity measure was sensitive to the visual feedback removal. Younger people are characterized by higher entropy values in both directions compared to older people when standing with both eyes open and closed [9,41]. As was suggested in paper [83,84], lower SampEn observed in older adults could have been due to utilization of greater muscle co-activation or joint rigidity as their postural strategy. Therefore, in this case of older adults lower SampEn represent more regular pattern for CoP variability, that is the rigid strategy, and lower adjustment to perturbations [85,86]. The comparison of healthy people with those suffering from chronic cervical pain, multiple sclerosis, cerebral palsy, chronic cervical pain following whiplash injury, or fibromyalgia [43,45,48,52,55] revealed that people with dysfunctions, injures, or diseases had lower values of entropy. Therefore, in this case the results of the entropy analysis suggest that people from this group need to concentrate more on postural control or else they lose complexity and automated postural control. Rigoldi, Cimolin, Camerota, Celletti, Albertini, Mainardi, and Galli [45] showed that patients with Ehlers–Danlos syndrome (EDSG) are characterized by a pronounced ligament laxity that, in the presence of hypotonia acts negatively on somatosensory postural control feedback, thus increasing the attentional demands of the visual and vestibular feedback systems. This translates into the less automated postural system, evidenced by a lower value of entropy. The last study group where the sample entropy ratio was used were athletes. The analyzed studies on athletes show that people involved in sports (mainly those involved in dancing, ballet, or gymnastics) show higher values of entropy than non-athletes [57,60]. Such results mean that athletes do not have to focus on performing stabilographic tasks, as they use a kind of automated movements [58]. At this point it is worth mentioning that several researcher have already proposed a relationship between entropy of the CoP signal and automaticity of sway [4,7,9,57,87]. Until now, two modes of postural control have been identified: (1) A controlled and ineffective mode and (2) an automatic mode [7,9,88]. Donker, Roerdink, Greven, and Beek [7] proofed that standing with eyes closed (i.e., creating an internal focus by increasing task difficulty through visual deprivation) significantly increased CoP regularity (indexed by a decrease in SampEn). Furthermore, variability increased and local stability decreased, implying ineffective postural control. Conversely, withdrawing attention from postural control (i.e., performing a cognitive dual task while standing with eyes closed) led to greater irregularity (increase of SampEn) and smaller variability, suggesting an increase in the ‘‘efficiency”, or ‘‘automaticity’’ of postural control’’. Therefore, Donker, Roerdink, Greven and Beek [7] showed that increased sway regularity (low SampEn) combined with decreased postural stability would indicate a controlled and ineffective postural control. While increased entropy combined with increased postural stability is representative of an automatized postural control, caused by experimentally withdrawing attention from the postural task.

The human body is continuously exposed to external perturbations, which can be counteracted by integrating the real-time inputs and the prediction system based on previous inputs. The information given by the nonlinear approach can well describe this mechanism. In a study by [44] the group of people suffering from chronic cervical pain after whiplash injury was characterized by lower FD than the control group. In both conditions of eyes open and closed, the values were lower for AP and ML directions. Qiu. H [66] showed that FD for the elderly has higher values in the conditions of eyes open. When comparing people from second group (Disabilities/Injures/Diseases) to healthy people, the fractal dimension (FD) differentiates the two groups in a precise manner. Biec, Zima, Wojtowicz, Wojciechowska-Maszkowska, Krecisz, and Kuczynski [46] demonstrated that FD is higher for sway in the sagittal plane in both tests with eyes closed and open for people with Down syndrome. Furthermore, Cimolin, Galli, Rigoldi, Grugni, Vismara, Mainardi, and Capodaglio [70] showed that compared to healthy people a higher fractal dimension is observed in patients with Prader–Willi syndrome (PWS). The higher FD value indicate a more complex and irregular signal over time. In case of the PWS population, this result may be influenced by functional profile of the PWS population, which is known to have poor balance and greater risk of fall than healthy individuals, caused by hypotonia and excessive body weight. On the other hand, the authors suggested that such a result may also be interpreted as an inability of those patients to synergistically modulate the three systems (i.e., the visual, vestibular, and somatosensory systems) involved in maintaining posture. In the group of athletes, ballet dancers obtained higher FD values than non-athletes [73]. Tassani, Font-Llagunes, Gonzalez Ballester, and Noailly [69] came to interesting conclusions, showing that everyday stressful situations can lead to reduced postural stability. In their studies, in the case of muscle group tension, the fractal dimension reached higher values than in the case of relaxed positions, both with eyes open and closed.

The application of the Lyapunov exponent in the research occurred the least frequently, which makes it the least common form of analysis of the CoP parameter. Ghofrani, Olyaei, Talebian, Bagheri, and Malmir [76] demonstrated that in healthy young people, the Lyapunov exponent reaches higher values than in older adults and increases in the absence of visual control. The results were also confirmed in the paper [41]. In the case of patients with multiple sclerosis (PWMS) [78] or people subjected to falls [74] it was shown that these people were characterized by lower Lyapunov exponent values in both directions compared to healthy people. Compared to healthy controls, movements of PWMS patients are less complex, more rigid as they use fewer movement strategies available to them. In conclusion, higher Lyapunov exponent values mean a healthy and highly irregular postural system that adapts to external perturbations/disturbances, and the lower the values, the higher the regularity and more reproducible patterns and the worse the adaptation to changing environmental conditions.

As shown in this review, the use of nonlinear dynamic coefficients provides insights into the specificity of the patient’s health status in terms of the amount of attention used to perform the balance task, the ability to adapt to possible destabilizing stimuli, the time of reaction to stimuli or adaptation to difficult conditions (dual-task, reduction of the support area, moving surface or the absence of visual control). In conclusion, it seems that an advantage of the nonlinear dynamics measures is that they can differentiate between subtle physiological changes such as ageing and balance-related neurological diseases. Moreover, nonlinear dynamics offer the assessment of the postural control system the ability to be adaptable and flexible in unpredictable and ever-changing environment. Furthermore, the literature review revealed that there have been no studies to make comprehensive use of all nonlinear dynamic measures, and these appear to complement or confirm the evaluation power.

## 5. Limitation of Study

The presented study has some limitations. The first limitations will result from the fact that some of the considered papers had limitations mentioned by the authors themselves. These limitations include: Small number of participants, no randomness in the test order, not evaluated lower-extremity muscle strength, which can significantly affect balance level in case of, e.g., Athletes group. The second group of limitations will result from the fact that in the case of sample entropy, the authors used different m and r coefficients for calculations, although the value of 2 and 0.2, respectively, was dominant. Moreover, the authors did not specify whether the SampEn was calculated on the raw or filtered signal. For fractal dimension calculation we can observe different methods of calculations: Higuchi’s or Katz’s algorithm. As an example of this review, it can be seen that nonlinear methods are beginning to be an attractive tool for assessing postural stability, but there are still no conservative guidelines how to calculate them in one way. The authors choose different parameters or different counting methods without further explanation in their articles.

## Figures and Tables

**Figure 1 entropy-22-01357-f001:**
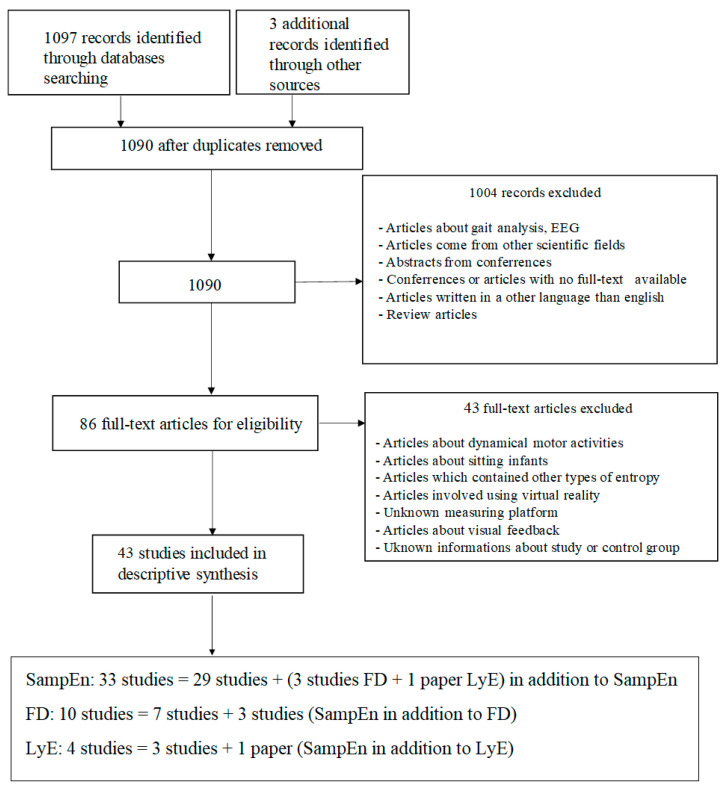
Flowchart demonstrating the selection of articles through the review process and subdivision into three groups, where: SampEn—sample entropy, FD—fractal dimension, LyE—Lyapunov exponent.

**Table 1 entropy-22-01357-t001:** Description of nonlinear measures calculated for center of pressure (CoP) time series signal.

Nonlinear Coefficients	Low Value	High Value
Sample entropy (SampEn)—a measure of the regularity and complexity of a signal and the amount of attention devoted to the performance of a given task. Values are comprised between 0 (perfectly regular sway) and 2 (totally irregular and unpredictable sway) [9].	1. Regular CoP time series. 2. Sign of possible pathology. 3. The system may not respond flexibly to a given destabilizing stimulus. 4. Rigidity for postural control. 5. System unable to successively adapt to new changes in the environment.	1. Irregular CoP time series. 2. Sign of a healthy, alert biological system. 3. System ready for the occurrence of an “unexpected” stimulus.
Fractal dimension (FD)—provides an indication of the complexity of a signal by analyzing the entire signal and describing its shape and may be indicative of a change of the control strategies used for upright balance [24].	A signal with a fractal dimension equal to 1 would indicate a completely stationary signal over time. An impossible situation in which a person stands completely still without swaying.	Randomly generated data or data with too high a noise component, then the fractal dimension converges to 2 [25].
Lyapunov exponent (LyE)—its positive value is considered a necessary and sufficient condition for the presence of chaos in the system. LyE provides a measure of the local stability of a dynamical system [7].	Indicates the rigidity of the system and the inability to adapt to the environment [26].	Indicates the ability to react faster to destabilizing stimuli and to better control the balance [26].

**Table 2 entropy-22-01357-t002:** Data extracted from reviewed articles for sample entropy, where: *—significant differences.

Study and Quality	Study Group	Age (Years)	Protocol/Conditions	Plate and Sampling Rate (Hz)	m, r and Fourth-Order Low Pass Butterworth Filter	Results/Findings
Group I: Children/Young/Older Adults						
[32] Quality: 10/11	Hypnotic susceptibility: 11 lows and 11 highs	22.9 ± 1.8 23.2 ± 2.4	4 trials (30s): E (easy—stable support); D (difficult—unstable support); B (basal, EC), MC (mental computation)	NI-DAG 6.9.3; 100 Hz	m = 2, r = 0.2 (no data)	SampEn_ML (Highs) (B/MC): E: 0.07 ± 0.04/0.08 ± 0.03 D: 0.10 ± 0.02/0.09 ± 0.02 SampEn_ML: D > E * Support x Task interaction: D > E * only during B. MC increased SampEn only in E *
[33] Quality: 10/11	Y: 14 OA: 11	23 ± 2 73 ± 6	2 trials standing (102.4 s) with EO and EC	Win- Posturo; 40 Hz	m = 3, r = 0.3 (no data)	Significant vision effect in: AP (EO/EC): Y: 1.091 ± 0.193/0.966 ± 0.158 * OA: 0.988 ± 0.243/0.905 ± 0.282 * ML (EO/EC): Y: 1.084 ± 0.213/0.961 ± 0.191 * OA: 0.964 ± 0.255/0.902 ± 0.282 *
[34] Quality: 10/11	Y right—handed: 22	24 ± 3.2	10 trials standing (30 s): 5 evenly distributed load and 5 unevenly distributed load. The specific loads held by the subjects were 1, 3, 5, 7, and 9 kg. Each trial was performed two times	AMTI AccuSway; 200 Hz	m = 2, r = 0.2 (6 Hz)	Significant weight × side interactions in SampEn_ML and AP: SampEn_left (loaded) limb < SampEn_right (unloaded) limb *. A Tukey post hoc: SampEn_AP was different at the 5, 7, and 9 kg loads. Only the 9 kg load was different for ML entropy. The resultant SampEn_ML tended to decrease with increasing load magnitude in the evenly and unevenly distributed load
[35] Quality: 10/11	Fallers/Non fallers: 30/45	74.4 ± 9.0	2 trials (60 s) with EO and EC	AMTI BP400600-2K; 1000 Hz	m = 3, r = 0.2 (1 Hz)	EO/EC: 0.52 ± 0.35/0.36 ± 0.24 EC for (Fallers/Non fallers): 0.33 ± 0.23/0.42 ± 0.24
[36] Quality: 10/11	AA: 10 FA: 15 LA: 8 NA: 5	76.3 ± 9.7 76.7 ± 8.0 81.9 ± 9.3 79.4 ± 7.0	1 trial standing (20 s) with EO	Kistler; 100 Hz	(no data)	AP/ML: AA: 0.93/0.73 FA: 0.79/0.52 LA: 0.79/0.68 NA: 0.78/0.62
[9] Quality: 10/11	Y: 21 OA: 25	22.5 ± 2.0 69.4 ± 3.4	Normal standing with EO and dual-task (2 discrete and 2 continuous). Each trial (60 s)	AMTI ORG-6-1000; 500 Hz	m = 2, r = 0.2 (no data)	SampEn_AP and ML: Y > OA
[37] Quality: 10/11	YO: 22 MO:37 OO: 31	65.4 ± 2.3 74.6 ± 2.7 85.4 ± 4.4	6 limits of stability trials: 3 on firm, 3 on foam pad	Bertec 5046; 100 Hz	m = 2, r = 0.2 (no data)	YO/MO/OO in firm plate: AP: 0.049 ± 0.018/0.070 ± 0.026/0.097 ± 0.040 ML: 0.021 ± 0.009/0.029 ± 0.015/0.039 ± 0.020 YO/MO/OO in foam plate: AP: 0.071 ± 0.017/0.092 ± 0.039/0.111 ± 0.040 ML: 0.031 ± 0.012/0.039 ± 0.018/0.047 ± 0.022
[38] Quality: 10/11	NP (Non-pregnant): 10 P1 (Pregnant I trimester): 10 P2: 10 P3: 10	23 (22–25) 28 (21–30) 24.5 (22.2–27) 25 (23.5–29.5)	Standing with EO (120 s)	Biomec; 100 Hz	m = 2, r = 0.2 (10 Hz)	AP/ML: NP: 0.09 (0.07–0.10)/0.14 (0.11–0.20) P1: 0.06 (0.06–0.07)/0.09 (0.08–0.13) P2: 0.07 (0.06–0.08)/0.08 (0.06–0.10) P3: 0.07 (0.05–0.07)/0.07 (0.05–0.07)
[39] Quality: 10/11	Y: 7	22.9 ± 1.1	10 trials (20 s): without and with the VFB (visual feedback)	AMTI AccuSway; 50 Hz	m = 2, r = 0.08 and 0.05 (signal estimated to 25 Hz)	SampEn_AP and ML: Y (VFB) > Y
[40]^FD^ Quality: 10/11	Y: 16	22–25	Quiet standing on a soft support surface with EO 4 times (20 s): before training, 1 min after, 30 min after, 24 h after	Kistler 9286AA; 100 Hz	m = 3, r = 0.02 (no data)	SampEn_ML > SampEn_ML_ 24h after training
[41]^LyE^ Quality: 10/11	Y: 15 OA: 15	22.1 ± 1.7 68.3 ± 2.7	4 trials (90 s): shoulder wide feet distance with EO and EC; narrow feet distance with EO and EC	AMTI OR6-6-1000; 1000 Hz	m = 2 and 3, r = 0.1, 0.15, 0.2, 0.25, 0.3 (data download sampled from 1000 Hz to 100 Hz)	Y, OA: SampEn_AP: EO < EC * OA: SampEn_ML: EO < EC *
[42] Quality: 10/11	3 y.o (years old): 16 4 y.o: 18 5 y.o: 23	3 years (42.3 ± 3.2 months) 4 years (52.4 ± 3.8 months), 5 years (65.3 ± 3.6 months)	4 trials standing (40 s): Standing on rigid surface with EO and EC; standing on a foam surface with EO and EC. For both EO conditions, the children were watching a movie	AMTI; 100 Hz	m = 3, r = 0.2 (12.5 Hz)	AP in EO/EC: 3.y.o: 0.79 ± 0.29/0.75 ± 0.24 4.y.o: 0.92 ± 0.25/0.79 ± 0.22 5 y.o: 0.62 ± 0.30/0.65 ± 0.25 ML in EO/EC: 3.y.o: 0.82 ± 0.38/0.78 ± 0.37 4.y.o: 0.83 ± 0.31/0.93 ± 0.34 5.y.o: 0.63 ± 0.30/0.60 ± 0.27 SampEn_AP_ML: main effect of age *, main effect of vision *, main effect of surface *
**Group II: Disabilities/Injures/Diseases**						
[43] Quality: 10/11	A: 11 CG: 13	10.3 ± 1.2 10.1 ± 1.3	3 tasks (20 s) each with EO and EC repeated 5 times: standing; standing on foam surface; standing while performing a cognitive DT	Custom made strain gauge force plate; 200 Hz	m = 3, r = 0.05 (no data)	SampEn: A < CG *. Task × Group interaction *: SampEn_foam < SampEn_other conditions Vision × Group interaction for CG *: SampEn: EC < EO
[44]^FD^ Quality: 10/11	CWJ: 11 CG: 11	33.3 ± 6.7 33.1 ± 6.8	3 trials standing (45s) with: EO, EC, EO and normal speaking (DT)	AMTI OR6-5-2000; 200 Hz	m = 2, r = 0.2 (10.5 Hz)	CWJ, CG: SampEn_EC_DT > SampEn_EO * SampEn_EC_DT: CG > CWJ *
[45] Quality: 10/11	EDSG: 13 CG: 20	32.4 ± 8.4 31.4 ± 9.6	1 trial (30 s) standing with EO and EC	Kistler; 500 Hz	(no data) 10 Hz	SampEn: EDSG < CG * (no differences between EO and EC) AP (EDSG/CG) *: EO: 0.05 ± 0.10/0.18 ± 0.20 EC: 0.05 ± 0.08/0.24 ± 0.27 ML (EDSG/CG) *: EO: 0.13 ± 0.16/0.29 ± 0.21 EC: 0.13 ± 0.19/0.39 ± 0.22
[46]^FD^ Quality: 10/11	DS: 10 CG: 11	29.8 ± 4.8 28.4 ± 3.9	4 trials standing (20 s) with: EO and EC on hard surface EO and EC on foam pad	Kistler 9286AA; 100 Hz	m = 3, r = 0.02 (no data)	Plane and surface significantly affected SampEn. AP (EO/EC) firm surface: DS: 0.75 ± 0.18/0.72 ± 0.16 CG: 0.97 ± 0.37/0.80 ± 0.27 ML (EO/EC) firm surface: DS: 0.69 ± 0.07/0.75 ± 0.05 CG: 0.65 ± 0.22/0.60 ± 0.14 AP (EO/EC) foam surface: DS: 0.67+0.10/0.65 ± 0.09 CG: 0.56+0.17/0.64 ± 0.09 ML (EO/EC) foam surface: DS: 0.68 ± 0.03/0.70 ± 0.02 CG: 0.58 ± 0.07/0.57 ± 0.05
[47] Quality: 10/11	ASD: 5 CG: 5	9.2 ± 0.45 7.4 ± 2.06	Postural stability evaluated pre- and post-intervention under 4 trials standing (20 s): Flat surface with EO and EC; foam surface with EO and EC	Bertec BP505; 100 Hz	(no data) 5 Hz	AP: CG/ASD pre intervention /ASD post intervention: EO: 0.13 ± 0.04/0.12 ± 0.04/0.08 ± 0.03 EC: 0.10 ± 0.03/0.12 ± 0.03/0.10 ± 0.03 ML: CG/ASD pre intervention /ASD post intervention: EO: 0.14 ± 0.04/0.09 ± 0.04/0.09 ± 0.04 EC: 0.13 ± 0.04/0.12 ± 0.04/0.12 ± 0.03
[48] Quality: 10/11	NP: 20 CG: 20	70.8 ± 4.1 71.4 ± 5.1	2 trials (30 s) standing with EO and EC	Kistler 9286A; 100 Hz	m = 3, r = 0.3 (no data)	SampEn: NP > CG, (NP/CG): EO: 1.72 ± 0.1/1.73 ± 0.1 EC: 1.66 ± 0.1/1.73 ± 0.1 *
[49] Quality: 10/11	CP: 30 CG: 30	8.30 ± 2.3 9.20 ± 1.9	6 trials (20s) standing in which no supra-postural task was performed. Next, they performed a supra-postural task requiring them to balance a marble inside a tube held in the hands	AMTI AccuSway; 100 Hz	m = 3, r = 0.2 (no data)	CP, CG (AP and ML): SampEn_task performance < SampEn_quiet-standing (AP and ML) SampEn_quiet-standing: CP > CG (AP and ML) SampEn_task performance: CP < CG
[50] Quality: 9/11	CP: 30 CG: 30	8.30 ± 2.3 9.20 ± 1.9	6 trials (20s) standing: easy and hard functional play task conditions were repeated 3 times	AMTI AccuSway; 100 Hz	m = 2, r = 0.2 (no data)	SampEn_quite-standing and task performance: CP > CG * (during task performance, differences were attenuated)
[51] Quality: 10/11	CP: 8 CG: 9	11 ± 3.3 9.4 ± 2.0	2 trials (30 s) standing with EO before and after a maximal aerobic shuttle-run test (SRT)	Bertec FP4060-08; 1000 Hz	m = 3, r = 0.05 (12 Hz)	SampEn_both the pre- and post-SRT tests CP < CG
[52] Quality: 9/11	FMG: 80 CG: 49	43–70 years	4 balance tasks (60s) repeated 2 times: standing with EO; DT with EO; standing with EC; standing on foam surface with EO; standing on foam surface with EC	Wii Balance Board; 40 Hz	m = 4, r = 0.35 (10 Hz)	SampEn_ ML (all tasks): FMG < CG *. AP (CG/FMG): EO: 0.082 ± 0.08)/0.77 ± 0.14 EC: 0.76 ± 0.10/0.67 ± 0.14 DT: 0.78 ± 0.09/0.71 ± 0.11 FEO: 0.70 ± 0.08/0.63 ± 0.12 FEC: 0.62 ± 0.07/0.54 ± 0.11 ML (CG/FMG): EO: 0.96 ± 0.06/0.92 ± 0.09 EC: 0.95 ± 0.06/0.88 ± 0.11 DT: 0.93 ± 0.07/0.90 ± 0.09 FEO: 0.83 ± 0.07/0.81 ± 0.09 FEC: 0.76 ± 0.07/0.71 ± 0.11
[53] Quality: 10/11	LAS: 18 CG: 12	66 ± 4.3 65 ± 4.0	3 trials (20s) of a single-leg standing with EO	AMTI AccuSway Plus; 100 Hz	m = 2, r = 0.2 (5 Hz)	LAS/CG: AP: 0.35 ± 0.16/0.42 ± 0.08 ML: 0.27 ± 0.12/0.37 ± 0.08
[54] Quality: 10/11	CAI: 22 LAS: 20 CG: 24	21.27 ± 4.59 21.65 ± 3.56 20.96 ± 2.10	3 trials (20s) of a single-leg standing on testing leg with EC	Bertec 4060NC; 100 Hz	m = 3, r = 0.3 (5 Hz)	SampEn_AP and ML: no significant differences between-group. SampEn_ML: CAI > LAS, CAI > CG
[55] Quality: 9/11	CG: 50 MS low: 34 MS mod: 27 MS high: 42	64.9 ± 4.9 54 ± 13.2 58.2 ± 8.3 56.7 ± 9.7	1 trial (30 s) standing with EO. In patients with MS risk of falls (low, moderate, high) was assessed using the short form of the Physiological Profile Assessment	Bertec FP4060-05-PT-1000; 1000 Hz	m = 3, r = 0.2 (10 Hz)	SampEn was identified as the strongest feature for classification of low-risk MS individuals from healthy CG
[56] Quality: 10/11	CAI: 19 CG: 16	22.32 ± 3.07 22.06 ± 3.75	1 trial (20 s) standing single-leg	AMTI OR6-5/kinematics analysis; 100 Hz	m = 2, r = 0.2 (6 Hz)	SampEn_AP and ML: CAI < CG
**Group III: Athletes**						
[57] Quality: 10/11	D: 14 CG: 16	11.5–13.3 11–13.2	Standing (20 s) with EO or EC and with or without performing an attention-demanding cognitive task (DT) (word memorization)	Custom made strain gauge force plate; 100 Hz	m = 3, r = 0.05 (12.5 Hz)	SampEn: D > CG, EO > EC, DT > normal trial
[58] Quality: 10/11	D: 33 CG: 22	20.3 ± 3.3 21.3 ± 2.3	2 trials standing (20 s): quite standing with EO and dual task (stroop test)	Kistler; 20 Hz	m = 2, r = 0.1 (no data)	ML (Single task/dual task): D: 0.87 ± 0.22/1.12 ± 0.24 CG: 0.85 ± 0.20/1.06 ± 0.25 AP (Single task/dual task): D: 0.75 ± 0.26/0.97 ± 0.38 CG: 0.87 ± 0.41/1.00 ± 0.31
[59] Quality: 10/11	B: 10 CG: 10	21.5 ± 3.1 21 ± 1.8	3 trials standing (20 s) on: two legs, one leg, toe standing (35 deg PF like in high heel)	AMTI; 100 Hz	m = 2, r = 0.2 (10 Hz)	(B and CG): SampEn_AP: standing both feet < one-leg standing *. A contrary trend for SampEn_ML was observed.
[60] Quality: 10/11	D: 18 CG: 30	23.3 ± 2.6 22.2 ± 1.8	2 trials standing (30s) with EO and EC	Lafayette 16020; 100 Hz	m = 2, 3, 4 r = 0.15, 0.2, 0.25 (no data)	ML (EO/EC): ND: 0.094 ± 0.030/0.082 ± 0.037 * D: 0.096 ± 0.028/0.058 ± 0.024 *
[61] Quality: 10/11	G: 10 CG: 10	21.9 ± 1.0 22.0 ± 1.3	3 trials (30 s) with EC	Dynatronic; 40 Hz	m = 3, r = 0.05 (5 Hz)	SampEn: G > CG *
[62] Quality: 10/11	D: 13 CG: 13	28.0 ± 7.0 23.0 ± 3.0	Quiet standing with EO and EC. LOS test—stand quietly during the first 10s (1st phase) next to lean as far (2nd phase) and as fast as they were able and then to maintain this position (3rd phase). Test are repeated three times and lasted 30 s	AMTI Accugait; 100 Hz	m = 2, r = 0.2 (7 Hz)	SampEn_EO and EC_quiet standing: D > CG. LOS_AP: D > CG (1st and 3rd phase)
[63] Quality: 10/11	D: 25 CG: 25	25.6 ± 3.8 24.7 ± 2.6	4 condition—unipedal standing balance tests (30 s): firm surface with EO and EC; foam surface with EO; and firm surface with EO immediately after performing ten 360˚ whole-body turns. (3 trials for each condition)	Kistler 9286AA; 200 Hz	m = 2, r = 0.15 (7 Hz)	SampEn_AP_EC: D > CG Group x condition interaction: significant for SampEn_AP. The effect of group was significant for ML and AP
	The minimum value obtained in the reviewed works: 0.021 ± 0.009 [37]				The maximum value obtained in the reviewed works: 1.73 ± 0.1 [48].		

Abbreviations: AP—anterior–posterior, ML—medial–lateral direction, EO—eyes open, EC—eyes closed, DT—dual task, Y—young, OA—older adults, AA—always active, FA—formerly active, LA—lately active, NA—never active, YO—Young-Old, MO—Middle-Old, OO—Old-Old, ASD—autism spectrum disorders, CAI—chronic ankle instability, MS—multiple sclerosis, LAS—lateral ankle sprain, CP—cerebral palsy, NP—neck pain, A—anxious children, EDSG—Ehlers–Danlos syndrome, FMG—fibromyalgia group, DS—Down syndrome, CG—control group, D—dancers, B—ballet group, G—expert gymnasts, CWJ—chronic whiplash injury, FD—study in Table 3, LyE—study in Table 4.

**Table 3 entropy-22-01357-t003:** Data extracted from reviewed articles for fractal dimension, where: *—significant differences.

Study and Quality	Study Group	Age (Years)	Protocol/Conditions	Plate and Sampling Rate (Hz)	Method of Calculation	Results/Findings
Group I: Children/Young/Older Adults						
[65] Quality: 10/11	Y: 18 OA: 26	23.8 ± 1.5 69.8 ± 5.6	4 trials (70 s) standing: (1) EO in the normal lighting conditions (NL), (2) EC in the normal lighting conditions (NL), (3) EO in a very low level of illumination (LL), (4) EO in complete darkness (D)	Kistler 9286AA; 50 Hz	Higuchi’s algorithm	FD for ML direction (Y/OA) over (0–0.3 s): (1) 1.09 ± 0.03/1.09 ± 0.02 (2) 1.10 ± 0.03/1.10 ± 0.04 (3) 1.09 ± 0.03/1.10 ± 0.04 * (4) 1.10 ± 0.03/1.10 ± 0.03 FD for AP direction (Y/OA) over (0–0.3 s): (1) 1.15 ± 0.05/1.13 ± 0.04 (2) 1.15 ± 0.06/1.14 ± 0.05 (3) 1.14 ± 0.04/1.13 ± 0.04 (4) 1.13 ± 0.05/1.13 ± 0.04
[66] Quality: 10/11	Y: 20 OA: 20	20.4 ± 1.8 69.4 ± 3.1	2 trials (20 s) standing with EO and EC. Each trial repeated 3 times	Pro Balance Master; 100 Hz	Katz’s algorithm [67,68]	EO: FD_Y > FD_OA * FD Reliability: high (ICC ≥ 0.75 and %SEM ≤ 10%) Insensitive to age group differences and fear of falling under both visual conditions
[40]^SampEn^ Quality: 10/11	Y: 16	22–25	Quiet standing on a soft support surface with EO, 4 times (20 s): before training, 1 min after, 30 min after, 24 h after	Kistler 9286AA; 100 Hz	Higuchi’s algorithm	Core stability exercises did not cause any changes in FD over time
[69] Quality: 10/11	Y: 30	31.0 ± 6.0	6 trials (60 s): Normal standing EO and EC, tense and relaxed states and with both EO and EC	AMTI Accu-gait; 100 Hz	Katz’s algorithm [67,68]	Conditions (EO/EC): Normal standing: 1.45 ± 0.07/1.48 ± 0.08 Relax: 1.44 ± 0.07/1.46 ± 0.07 Tense: 1.51 ± 0.08/1.55 ± 0.09
**Group II: Disabilities/Injures/Diseases**						
[70] Quality: 9/11	PWS: 11 CG: 20	34.4 ± 3.7 31.4 ± 9.6	Normal standing EO (30 s)	Kistler; 500 Hz	The box-counting method [71]	PWS/CG: 1.58 ± 0.08/1.12 ± 0.08 *
[44]^SampEn^ Quality: 10/11	CWJ: 11 CG: 11	33.3 ± 6.7 33.1 ± 6.8	3 trials standing (45 s) with: EO, EC, EO and normal speaking (DT)	AMTI OR6-5-2000; 200 Hz	Fractal dimension by dispersion analysis based on the standard deviation [72]	All trials FD: CWJ < CG DT task, FD_AP and ML: CWJ < CG * AP: CWJ/CG: 1.06 ± 0.01/1.14 ± 0.01 * ML: CWJ/CG: 1.21 ± 0.01/1.28 ± 0.01 *
[46]^SampEn^ Quality: 10/11	DS: 10 CG: 11	29.8 ± 4.8 28.4 ± 3.9	4 trials standing (20s) with: EO and EC on hard surface EO and EC on foam pad	Kistler 9286AA; 100 Hz	Higuchi’s algorithm	Hard surface AP (DS/CG): EO: 1.46 ± 0.10/1.38 ± 0.08 EC: 1.49 ± 0.08/1.41 ± 0.07 Hard surface ML (DS/CG): EO: 1.41 ± 0.07/1.42 ± 0.06 EC: 1.47 ± 0.05/1.42 ± 0.06 Foam pad AP (DS/CG): EO: 1.53 ± 0.09/1.39 ± 0.08 EC: 1.55 ± 0.08/1.44 ± 0.05 Foam pad ML (DS/CG): EO: 1.52 ± 0.05/1.52 ± 0.06 EC: 1.45 ± 0.06/1.46 ± 0.025
[15] Quality: 10/11	FIG: 29 SIG: 28 CG: 16	23.2 ± 4.3 21.5 ± 3.3 22.4 ± 1.7	3 trials standing (20 s) with EC: SIG—group with acute ankle sprain successfully completed the task on their non-injured limb; FIG—group with acute ankle sprain failed to complete their attempt on their injured limb; CG—group with no current injury successfully completed the task on their non-dominant limb	AMTI; 100 Hz	Katz’s algorithm [67,68]	FD: FIG < SIG < CG SIG: 1.58 ± 0.06 FIG: 1.54 ± 0.07 CG: 1.64 ± 0.06
[14] Quality: 10/11	LAS: 66 CG: 19	23.2 ± 4.9 22.5 ± 1.6	3 trials standing on single limb (20s) with EO and EC	AMTI; 100 Hz	Katz’s algorithm [67,68]	LAS/CG for involved limb: EO: 1.18 ± 0.14/1.21 ± 0.13 EC: 1.25 ± 0.14/1.39 ± 0.16 LAS/CG for uninvolved limb: EO: 1.15 ± 0.14/1.13 ± 0.15 EC: 1.23 ± 0.14/1.37 ± 0.21
**Group III: Athletes**						
[73] Quality: 10/11	BD: 10 CG: 10	23.7 ± 2.5 27.6 ± 3.5	5 × 30 s	Kistler 9286B; 100 Hz	Katz’s algorithm [67,68]	BD: 1.76 ± 0.06 CG: 1.68 ± 0.07
The minimum value obtained in the reviewed works: 1.06 ± 0.01 [44].				The maximum value obtained in the reviewed works: 1.76 ± 0.06 [73].		

Abbreviations: EO—eyes open, EC—eyes closed, AP—anterior–posterior, ML—medial–lateral direction, DT—dual task, Y—young people, OA—older adults, CG—control group, PWS—Prader–Willi syndrome, CWJ—chronic whiplash injury, DS—Down syndrome, LAS—lateral ankle sprain, BD—ballet dancers, FD—fractal dimension, SampEn—study in Table 2, LyE—study in Table 4.

**Table 4 entropy-22-01357-t004:** Data extracted from reviewed articles on the Lyapunov exponent, where: *—significant differences.

Study and Quality	Study Group	Age (Years)	Protocol/Conditions	Plate and Sampling Rate (Hz)	Method of Calculation	Results/Findings
Group I: Children/Young/Older Adults						
[74] Quality: 10/11	OA: 16 Y: 16	65.7 ± 6.1 25.7 ± 3.1	2 trials (100 s) standing with EO and EC	AMTI OPT400600 and electrical goniometer; 1000 Hz	The multivariate largest Lyapunov exponent and the Wolf’s method [75]	LyE: Y < OA. LyE has a high accuracy to distinguish subjects from different groups under the EC conditions
[76] Quality: 9/11	M: 7 F: 8	22.2 ± 0.9 24.0 ± 2.8	3 trials standing (30 s) in each condition: EO and EC	Bertec; 500 Hz	The Chaos Data Analyzer software [77], based on the Wolf’s method [75]	Trial 1/Trial 2/Trial 3 EO: 0.12 ± 0.07/0.13 ± 0.09/0.13 ± 0.09 EC: 1.86 ± 0.53/1.80 ± 0.89/2.23 ± 0.67
[41]^SampEn^ Quality: 10/11	Y: 15 OA: 15	22.1 ± 1.7 68.3 ± 2.7	4 trials (90 s): shoulder wide feet distance with EO and EC; narrow feet distance with EO and EC	AMTI OR6-6-1000; 1000 Hz	Wolf’s method [75]	AP: LyE_OA > LyE_Y for all trials OA: LyE_EC > LyE_EO for all trials * Y: LyE_wide feet > LyE_narrow *
**Group II: Disabilities/Injures/Diseases**						
[78] Quality: 10/11	PWMS: 15 CG: 15	45.1 ± 10.5 39.4 ± 11.7	5 min standing with EO 5 min standing with EC	Kistler 9281-B11; 100 Hz	The Chaos Data Analyzer software [77], based on the Wolf’s method [75]	LyE in ML: PWMS < CG LyE in AP: PWMS < CG *
The minimum value obtained in the reviewed works: 0.12 ± 0.07 [76]				The maximum value obtained in the reviewed works: 2.23 ± 0.67 [76]		

Abbreviations: EO—eyes open, EC—eyes closed, DT—dual task, AP—anterior–posterior, ML—medial–lateral direction, Y—young, OA—older adults, M—males, F—females, CG—control group, PWMS—patients with multiple sclerosis, SampEn—study in Table 2.

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
