# Peer review of "Nonlinear Measures to Evaluate Upright Postural Stability: A Systematic Review"

_entropy, 2020, doi:10.3390/e22121357_

Round 1
Reviewer 1 Report
This contribution presents a thorough and comprehensive review of non-linear methods to assess postural stability. The chosen methods are Sample entropy (SampEn), Lyapunov exponent, and fractal dimension. The authors searched several databases according to selected keywords, visually inspected the references included in the found papers, and, with additional exclusion criteria, found the papers that match their aim. This is a standard procedure for writing such a review, and the authors followed the usual pattern. Such reviews, although without new results, are valuable for researchers that wish to work on similar topics, as a quick survey of achievements of other researchers can be a good guideline for their work.
The paper is well written and easy to follow. All the main conclusions of the analyzed papers are written and discussed.
However, there are some very minor comments.
Line 105: parameters m and r are mentioned. Only the researchers familiar with ApEn and SampEn know what they are. For the others, they should be defined, especially as they are used further on.
Line 66: The creators of SampEn [10] did state that it is independent of time series length. However, although SampEn is considerably less sensitive to time series length than ApEn, from the papers that made the “length profile’’ of entropy it can be seen that some dependence on time series length exists.
Line 193: Section title and the remaining text are on different pages.
Table 1: It is interesting to see that some authors use the values < 0.1 for threshold. Threshold profiles can be found in many papers and they show that, although SampEn is generally insensitive to the threshold, an exception is r < 0.1, where the standard deviation of estimated entropies is large. However, it has nothing to do with this paper.
It was a pleasure to review a well-written paper.
Reviewer 2 Report
In this article, the authors present the results of a bibliographical search related to the application of established non-linear indicators for order and chaos in dynamical systems to assess postural stability during upright standing. Particularly, they review articles that refer to sample entropy (samplEn), fractal dimension (FD), and maximum Lyapunov exponent (LyE) in time series produced from the center of pressure signals (CoP).
Of 1100 related to the subject articles 43 were studied, from 33 articles mainly referring to sample entropy, 10 articles mainly on fractal dimension, and 4 papers on the Lyapunov exponent. After defining specific classification criteria the results of the selected papers are tabulated into groups and subgroups according to I) Age (Children/ Young/ Older adults), II) Disabilities/ Injures/ Diseases and III) Athletes, and presented in 3 tables, each for the non-linear coefficients. Finally, the authors draw a few conclusions as far as the applicability and usefulness of the non-linear dynamical indicators in assessing postural stability are concerned.
The paper is well written in good English and presented in logical order. The only remark I have to make is Table 4, which I would put at the beginning of the paper. In this table, the authors wisely explain the meaning of the non-linear measures and what low and high values signify in interpreting the CoP signals.
The article admittedly presents a careful bibliographical work. However, the number of the articles found with respect to the questions addressed is limited and can not validate the conclusions of the authors. In my opinion, the article includes preliminary work that should be done for the research on the applicability of non-linear coefficients on postural stability, and thus, I can not suggest it for publication.
